# Navigating the Semiochemical Landscape: Attraction of Subcortical Beetle Communities to Bark Beetle Pheromones, Fungal and Host Tree Volatiles

**DOI:** 10.3390/insects16010057

**Published:** 2025-01-09

**Authors:** Leah Crandall, Rashaduz Zaman, Marnie Duthie-Holt, Wade Jarvis, Nadir Erbilgin

**Affiliations:** 1Department of Renewable Resources, University of Alberta, Edmonton, AB T6G 2H1, Canada; rashaduz@ualberta.ca (R.Z.); erbilgin@ualberta.ca (N.E.); 2Ministry of Forests, Government of British Columbia, Cranbrook, BC V1C 7G1, Canada; Marnie.DuthieHolt@gov.bc.ca (M.D.-H.); wade.jarvis@gov.bc.ca (W.J.)

**Keywords:** *Ophiostoma montium*, *Leptographium longiclavatum*, *Grosmannia clavigera*, *Atropellis piniphila*, *Endocronartium harknessii*, Cerambycidae, Buprestidae, Elateridae, Staphylinidae, volatile organic compounds (VOCs), forest health

## Abstract

We investigated how subcortical beetle communities, including bark beetles, wood-boring beetles, and their predators, show attraction to different semiochemicals (chemical signals) from stressed pine trees, fungi, and a bark beetle pheromone in the field. Our findings indicate that fungal volatiles may enhance mountain pine beetle attraction to its commercially available lures. Catches of predatory beetles were associated with host volatiles, specifically, the volatile profile of healthy host trees of their prey. The response of bark beetle competitors to fungal and host stress volatiles was found to change in the presence of the mountain pine beetle lures. These findings suggest that fungal and host stress volatiles may be useful semiochemicals for monitoring and managing subcortical beetle populations.

## 1. Introduction

Subcortical coleopteran beetles, including bark beetles (Curculionidae, Scolytinae), long-horn (Cerambycidae) and short-horn (Buprestidae) beetles, and predators (Elateridae and Staphylinidae), develop and feed within the subcortical tissues (phloem and sapwood) of host trees. These beetles interact with a wide range of semiochemicals to locate suitable habitats, including volatiles released by host and non-host trees, fungal pathogens, and bark beetle pheromones [1,2,3]. However, these interactions can be complex as there are multiple sources of semiochemicals, each showing its unique properties [1,2,4,5,6]. We currently have little understanding of the semiochemical landscape governing the attraction of these habitat-sharing beetle communities.

Semiochemicals are known to be effective and eco-friendly tools that have been employed for over 50 years to monitor forest insect pest populations and gain a better understanding of insect diversity [6,7,8,9]. Species-specific pheromones are commonly used to monitor economically important bark beetle species [1,6,10]. Competitors, predators, and other bark beetle species that share the same subcortical habitats can also use bark beetle pheromones and host tree volatiles as cues to locate habitats used by their wide range of prey species [10,11,12,13]. Furthermore, host trees stressed by abiotic and biotic factors release different compositions of volatile chemicals, called volatile organic compounds or VOCs, from healthy trees [14,15,16]. These volatiles can act as attractants, repellents, or synergists to bark beetle pheromones [2,17,18]. More recently, studies explored the volatile compounds emitted by fungal associates of bark beetles as an additional source of semiochemicals influencing the subcortical beetle communities [19,20,21,22]. In contrast to the bark beetle pheromones, our understanding of the attraction of subcortical beetles to host stress volatiles and fungal volatiles is lacking.

Subcortical beetle interactions with their preferred semiochemicals differ between species. Table 1 summarizes the known attractants for various subcortical beetle communities. In general, ethanol and some monoterpenes released by host trees are attractive to some cerambycid species [23]. Traps combining tree stress volatiles are the most effective for some buprestid species [24,25]. To our knowledge, no long-range pheromones have been identified for buprestids. Predators of bark and wood-boring beetles, such as species from the Elateridae and Staphylinidae families, use bark beetle pheromones to locate host trees under attack by bark beetles [12]. For example, pheromones produced by several species of *Ips* (ipsenol and ipsdienol) are known to attract several wood-boring and predatory beetle species [26]. However, most bark beetle predators are habitat specialists– they predate on several species of beetles in the same subcortical habitat—and respond to a wide range of semiochemicals [12,27]. For instance, host volatiles are also known to be sources of attraction of bark beetle predators and can alter predator response to bark beetle pheromones [28,29].

We investigated the subcortical semiochemical landscape in the mountain pine beetle [MPB, *Dendroctonus ponderosae* (Hopkins)] system. The MPB is an eruptive species native to conifer forests in western North America [56,57,58,59]. For MPB, host tree colonization is primarily driven by the release of its aggregation pheromones, *trans*-verbenol and *exo*-brevicomin. Host tree volatiles, terpinolene and myrcene, are known to improve MPB attraction to its pheromones [33]. This four-component lure is commercially available to monitor MPB populations in western North America.

A large diversity of competitors and predators exist throughout MPB ranges. Major insect predators of MPB include species from Cleridae, Trogossitidae, Staphylinidae, Histeridae, Elateridae, and Dolichopodidae families. In particular, staphylinid and clerid beetles are important predators due to their ability to predate on multiple life stages of MPB [57,60,61]. Competitors include other bark and wood-boring beetle species, such as species from the Buprestidae and Cerambycidae families and the Scolytinae subfamily [61,62].

To successfully survive and reproduce within their selected host, MPBs depend on symbiotic fungi, *Grosmannia clavigera*, *Ophiostoma montium*, and *Leptographium longiclavatum*. These fungi help beetles overwhelm host defences during host colonization and provide essential nutrients such as nitrogen to immature stages of beetles post-host colonization [63,64,65,66,67]. Different concentrations and compositions of fungal volatile organic compounds (FVOCs) are released by different fungal species [68]. These FVOCs could be particularly important in host location by subcortical beetle communities [19,68]. For instance, a blend of volatiles released by ophiostomatoid fungi increased the attraction of *Ips typographus* to its aggregation pheromones in field trials [21].

In the MPB system, Zaman et al. (2023b) [22] recently characterized the FVOCs of the three main symbiotic fungi of MPB and identified the most abundant FVOCs released by each species. The FVOCs produced by all three fungi were similar and included 2-methyl-2-butanol, acetoin, isobutanol, 3-methyl-1-butanol, and 2-methyl-1-butanol. However, the concentrations of these chemicals varied; i.e., *L. longiclavatum* had the highest concentrations of isobutanol, 3-methyl-1-butanol, 2-methyl-1-butanol, and acetoin. Olfactometer bioassays in the same study also showed MPB attraction to FVOCs emitted by *G. clavigera* and *O. montium*. However, the effectiveness of these compounds as attractants for MPB and other subcortical beetles has not yet been tested in the field.

Host stress volatile compounds (SVCs) are another important source of semiochemicals for subcortical beetle communities [30]. These SVCs are released by trees in response to the presence of biotic and abiotic stressors [15]. The two most common SVC groups are green leaf volatiles (alcohols, acetates, and aldehydes) and volatile monoterpenoids and sesquiterpenoids [69]. Zaman et al. (2023a) [16] characterized the stress volatile profiles of lodgepole pine (*P. contorta* Dougl. ex Loud. var. *latifolia* Engelm) trees inoculated with three species of MPB symbiotic fungi and two common phytopathogens, *Atropellis piniphila* and *Endocronartium harknessii*. The most abundant monoterpenoids were β-phellandrene, 3-carene, α-pinene, and limonene. Terpene concentrations varied among the three symbiont fungal species; however, the attraction of subcortical beetles, including MPBs, to these SVC profiles is unknown.

Hence, the primary objective of this study is to examine how SVCs and FVOCs impact the attraction of beetle communities colonizing the same subcortical habitat as MPB. Our objectives were to determine whether (1) individual FVOCs and SVCs are attractive to subcortical beetle communities and (2) FVOCs synergize the attraction of MPB to its pheromones. Gaining a better understanding of the interactions between these beetles and their semiochemical landscape will allow managers and researchers to better understand subcortical beetle community dynamics, monitor population trends, and implement timely control measures to mitigate bark beetle populations.

## 2. Materials and Methods

### 2.1. Experimental Design

Three experiments were conducted simultaneously. In the first experiment, we tested the effects of FVOCs alone on subcortical beetle attraction. Our second experiment tested the potential synergistic effects of FVOCs and MPB pheromones. In the third experiment, we tested the effects of SVCs alone on subcortical beetle attraction. All three experiments were conducted in three blocks at two sites located approximately 20 km south of Cranbrook (British Columbia, Canada) (49°23′42.2″ N 115°34′27.6″ W and 49°23′33.2″ N 115°34′59.7″ W) in 2023 (Figure 1a). All sites were located in open, clear-cut areas surrounded by lodgepole pine forests, which were experiencing low-density MPB attacks during the experiment. A randomized block design was used with a 10 m distance between traps within a block and blocks at least 250 m apart (Figure 1b,c). The two sites were located 640 m apart and had similar slopes and aspects. Traps at the first site included dispensers containing FVOCs with MPB lure. These traps were placed at a different location from the other two experiments to avoid possible spillover effects from using the MPB lure. Traps at our second site tested the effect of FVOCs alone or SVCs alone in different blocks.

In all experiments, traps were suspended from 2 m-long conduits (Figure 1d). Flight intercept traps were used for the FVOC experiments (alone or with the MPB lure), and Lindgren traps for the SVC experiment. Insecticide strips were placed inside the collection cups for dry-cup collections. Volatile chemicals were combined with a mineral oil solvent in 15 mL, thin-walled, all-polyethylene release bottles and hung on the inside of traps (Figure 1d). Mineral oil was used as a solvent since it is not volatile does not react with the volatile chemicals or the plastic dispensers used, and allows for a controlled release of volatiles. All the volatile chemicals used in this experiment were purchased from commercial vendors (Table 2). The concentrations of FVOCs and SVCs were based on the concentrations reported by Zaman et al. (2023a & 2023b) [16,22]. Trap collections were completed every week for five weeks during the MPB flight period, starting on 13 July and finishing on 10 August, yielding 15 replications for each treatment. Between the first and second collections, a large wind event knocked down the flight intercept traps; therefore, the second collection was excluded from all data analyses. Rainwater was removed from sampling cups in the field using a fine mesh filter. After each collection, the samples were stored in a −20 °C freezer before identification. We provide further details regarding the volatiles tested below.

#### 2.1.1. Testing the Effects of FVOCs on the Attraction of Subcortical Beetles

Zaman et al. (2023b) [22] quantified the abundance of FVOCs associated with three species of MPB symbiotic fungi by inoculating lodgepole pine bolts with *G. clavigera*, *O. montium*, *L. longiclavatum* and a control (fungus-free) treatment. After 14 days, volatiles were extracted from infected regions of the phloem for each species and analyzed using a Gas-Chromatograph-Mass-Spectrometer (GC-MS). Although up to 60 volatile compounds, including monoterpenes, sesquiterpenes, diterpenes, and oxygenated monoterpenes, were detected and quantified, the compounds selected for this study were the most abundant. They represented nearly 90% of the total emissions for each fungal species, with the remaining compounds constituting less than 10% of the total concentration. The five most abundant volatiles identified by Zaman et al. (2023b) [22] that were used in our experiments were isobutanol, 2-methyl-2-butanol, 3-methyl-1-butanol, 2-methyl-1-butanol, and acetoin. Our treatments also included a blend of FVOCs based on the volatile profile of *O. montium* and blank control (mineral oil). The *O. montium* volatile profile was used due to the species being dominant in hotter, drier conditions, which is characteristic of our field site locations in late spring and summer [70]. We tested seven FVOC treatments (five individual volatiles, a blend, and a control) in three blocks for 21 traps. Table 2 contains treatment information and the individual semiochemicals used in this experiment.

#### 2.1.2. Testing the Combined Effects of FVOCs and MPB Pheromones on Subcortical Beetle Attraction

Traps that contained FVOCs with MPB lures included the same volatile treatments (the five FVOCs and a blend treatment) with MPB lures added. We used MPB lures alone as the positive control. The MPB lure was a four-component lure containing *exo*-brevicomin, *trans*-verbenol, terpinolene, and myrcene (Synergy Semiochemicals lure #3093) at a ratio of 2:2:1:1. The average release rates of the *exo*-brevicomin and *trans*-verbenol components were 1.71 mg/day and 0.23 mg/day, respectively, at 30 °C. We tested seven treatments in three blocks for 21 traps.

#### 2.1.3. Testing the Effects of SVCs on the Attraction of Subcortical Beetles

Zaman et al. (2023a) [16] characterized the SVC profiles of lodgepole pine associated with five phytopathogens: the three main MPB symbiotic fungal species (*O. montium*, *L. longiclavatum*, and *G. clavigera*), two other phytopathogens infecting lodgepole pine (*A. piniphila* and *E. harknessii*) and the profile of healthy trees (control). Including the SVC profiles associated with *A. piniphila* and *E. harknessii* allowed a comparison of subcortical beetle attraction to SVC profiles of (a) pathogens directly associated with the presence of MPB in host trees and (b) other common lodgepole pine phytopathogens. Phloem samples were collected from mature lodgepole pines infected by *A. piniphila* and *E. harknessii*. Healthy lodgepole pines were also inoculated with plugs of MPB symbiotic fungi and a control (fungus-free) agar plug. Phloem samples were collected from the trees at regular intervals. Two weeks after inoculation, the concentrations of monoterpenes and oxygenated monoterpenes were the highest. Therefore, these data were used for the analysis. Volatiles were extracted from ground phloem tissues and analyzed in GC-MS [16]. The SVC profiles of these five phytopathogens were composed of geranyl acetate, α-pinene, camphene, β-pinene, 3-carene, β-myrcene, limonene, terpinolene, bornyl acetate, γ-terpinene, camphor, and borneol mixed at different concentrations depending on the stress agent (Table 3, [16]). The average release rates of each SVC profile blend were determined by weight lost in the laboratory at 22 °C; release rates were similar across all SVC profiles (22.06 mg/day). Volumes of the blends added to each dispenser were the amount needed to last for 5 weeks in the field based on the release rates. We tested seven treatments: SVC profiles of three symbiotic fungi, two other common phytopathogens, a healthy tree profile (positive control) and mineral oil (negative control) in three blocks for 21 traps.

### 2.2. Statistical Analysis

All the subcortical beetles collected belonged to five coleopteran families and one subfamily, including Staphylinidae, Elateridae, Cerambycidae, Buprestidae, and Curculionidae. Notably, all catches of Curculionidae were species from the Scolytinae subfamily and were almost entirely MPBs. R-studio (version 2022 4.2.1) was used to perform all statistical analyses. To determine the relative attractiveness of the FVOCs and SVCs, we used the percent relative catches of beetles caught in each treatment throughout collections 1, 3, 4, and 5. Key treatments for the subcortical beetle attraction were identified by comparing the relative catches from each volatile treatment to the control treatments within the same experiment.

To understand the temporal variation of subcortical beetles caught throughout the field season, we compared the mean number of beetles caught during our four collection periods. We then conducted a Kruskal–Wallis rank-based test to determine whether there were significant differences between catches in collection periods for each subcortical beetle family. The Kruskal–Wallis test was used because the data for each beetle group exhibited non-normal distributions and had similar distributions across groups. For beetle groups found to have significant differences between collection periods, we conducted Dunn’s test for pairwise comparisons using Bonferroni’s *p*-value adjustment. We performed the same analyses using Kruskal–Wallis and Dunn’s tests to determine significant differences between catches of subcortical beetles between all the volatile treatment groups tested in our experiments. A non-metric multidimensional scaling (NMDS) ordination was also performed to understand differences in community compositions of subcortical beetles between volatile treatment groups, and a PERMANOVA test with pairwise comparisons was used to test for significance (*p* < 0.05). Due to the non-normal distribution and zero-inflated nature of the count data, the Bray–Curtis distance measure was used for our ordinations.

## 3. Results

We caught 1382 subcortical beetles, including 724 scolytids, 436 cerambycids, 96 elaterids, 92 buprestids, and 34 staphylinids throughout the field season (Table 4). Of the 724 scolytids caught in our traps, only 14 were not MPBs.

### 3.1. The Effects of FVOCs, SVCs and MPB Lure on the Attraction of Subcortical Beetles

We found no statistical differences between the volatile treatments used in all three experiments for any of the subcortical beetle groups. FVOCs alone: Cerambycidae (*p* = 0.936, df = 6), Buprestidae (*p* = 0.597, df = 6), Elateridae (*p* = 0.855, df = 6), and Staphylinidae (*p* = 0.863, df = 6); FVOCs with pheromone lure: Scolytinae (*p* = 0.896, df = 6), Cerambycidae (*p* = 0.636, df = 6), Buprestidae (*p* = 0.987, df = 6), Elateridae (*p* = 0.565, df = 6), and Staphylinidae (*p* = 0.613, df = 6); SVCs alone: Scolytinae (*p* = 0.936, df = 6), Cerambycidae (*p* = 0.793, df = 6), Buprestidae (*p* = 0.417, df = 6), Elateridae (*p* = 0.244, df = 6), and Staphylinidae (*p* = 0.533, df = 6).

#### 3.1.1. Catches of Subcortical Beetles in FVOC Treatments

For the two families of MPB competitors, cerambycids and buprestids, 2-methyl-1-butanol caught 21% and 36% of beetles, respectively (Appendix A). Of the total cerambycids caught, 18% were caught in the 3-methyl-1-butanol treatment compared to 11% in the control. Relatively low numbers of bark beetles (<5 individuals) were caught in FVOC-alone treatments. The isobutanol and FVOC blend each caught 22% of staphylinid beetles (Appendix A). The control treatment caught approximately 11% of staphylinid beetles. We caught very few elaterid beetles (18 total), and the 2-methyl-1-butanol, 2-methyl-2-butanol, acetoin, the blend, and control treatments each caught 17% of the total elaterids (Appendix A).

#### 3.1.2. Catches of Subcortical Insects in FVOCs with MPB Lure

In traps containing both FVOCs and MPB pheromones, 21% of scolytid beetles were caught in the 2-methyl-1-butanol, 17% in the 2-methyl-2-butanol, and 11% in the control (Appendix A). For cerambycid beetles, the percent catches of each FVOC changed with the addition of MPB lures: the acetoin (32%) and the FVOC blend (25%) accounted for over 50% of all trap catches (Appendix A). The 2-methyl-1-butanol (10%) and 3-methyl-1-butanol (4%) accounted for less than 15% of cerambycids caught. For buprestids, isobutanol caught 24%, and the FVOC blend caught 18% of beetles (Appendix A). Meanwhile, all the other treatments (including the control) each caught approximately 12% of the buprestid beetles. A quarter of all staphylinids caught were in the acetoin treatment, and 17% were in the 3-methyl-1-butanol treatment (Appendix A). No staphylinids were caught in traps containing the FVOC blend. A quarter of elaterid beetles were caught in the 2-methyl-2-butanol treatment (Appendix A). The 3-methyl-1-butanol, the blend, and the control treatments each caught 18% of elaterids.

#### 3.1.3. Catches of Subcortical Insects in SVC Treatments

Very few Scolytinae were caught in the SVC treatments (30 total). Therefore, key SVC profiles in MPB attraction were difficult to identify (Appendix A). Overall, the *L. longiclavatum* profile caught 33% of buprestids and 17% of cerambycids (Appendix A). Over 20% of the cerambycids caught were in the *E. harknessii* treatment. Approximately half of the staphylinids caught were found in the *G. clavigera* (24%) and healthy tree (21%) profile treatments (Appendix A). The healthy tree treatment also caught 30% of elaterid beetles (Appendix A).

### 3.2. Temporal Variation in Subcortical Beetle Catches

Except for Buprestidae (χ^2^_4_ = 4.679, *p* = 0.197), we found significant differences between collections for all subcortical beetle families: Scolytinae (χ^2^_4_ = 12.473, *p* = 0.006), Cerambycidae (χ^2^_4_ = 51.295, *p* < 0.001), Elateridae (χ^2^_4_ = 12.151, *p* = 0.007), and Staphylinidae (χ^2^_4_ = 60.397, *p* < 0.001). Using pooled data from all three experiments, we observed the highest mean catches of Scolytinae at the beginning (13 July) and end (10 August) of our sampling period (Figure 2a), which is likely due to two separate MPB emergence periods occurring during the field season. We caught very low numbers of MPBs in our third (27 July) and fourth (3 August) collections. We observed similar trends for the other subcortical beetle families (Figure 2d,e). Interestingly, the temporal variation in Elateridae and Scolytinae catches was very similar; mean catches during the first collection were significantly higher than collections 3 and 4. The trends of Cerambycidae and Staphylinidae catches throughout the field season were also very similar, with the lowest mean catches during collections 1, 3 and 4 and the highest in the fifth collection (Figure 2c,d).

### 3.3. Differences in Subcortical Beetle Catches Between Volatile Groups

The NMDS analysis, combined with a PERMANOVA and pairwise comparisons, revealed significant differences in subcortical beetle community composition among traps baited with different volatile groups (Figure 3). Volatile treatment groups accounted for 21.26% of the total variation in community composition. Traps treated with fungal volatiles combined with pheromones (FVOCP) exhibited significant differences in beetle composition compared to the mineral oil control, FVOC alone, healthy tree, and SVC treatments (*p* = 0.0019). Similarly, the pheromone-alone control group differed significantly from the mineral oil control, FVOC, healthy tree, and SVC treatments (*p* = 0.002). No significant differences in subcortical beetle community composition were found between volatile groups and their respective control treatments; for example, the FVOCP and pheromone-alone (control) volatile groups (*p* = 0.947).

In the NMDS plot, we observe clustering of the FVOCP and pheromone-alone (control) groups, which are spatially separate from the other volatile groups and are associated with increased catches of buprestids, scolytids, and elaterids. The FVOCP experiment accounted for 86% of scolytid and 44% of buprestid catches, compared to 10% and 6% in the pheromone-alone control group, respectively (Figure 4a,b).

Significant differences were also observed between the FVOC and SVC volatile groups (*p* = 0.01). Relative catches of staphylinidae in the SVC group were found to be 43%, compared to 16% in the healthy tree and 12% in the mineral oil control groups. Similarly, 30% of elaterids were caught in the SVC group, approximately triple that of the mineral oil control group (9%). Cerambycids were the only group of subcortical beetles with the highest relative catches in the FVOC group (35%) compared to other volatile groups, including FVOCP (28%), SVC (21%), and the mineral oil control (8%). No statistical differences were found between volatile treatment groups when comparing mean catches of individual insect families using the Kruskal–Wallis rank-based test.

## 4. Discussion

Our findings show that subcortical beetle communities interact with MPB lure, FVOCs, and SVCs differently; the community composition of subcortical beetle catches in our traps differed significantly among volatile groups. We identified FVOCs and MPB lure as important sources of semiochemicals due to their potential synergistic effects on MPB attraction. On the other hand, SVCs played a role in the attraction of predatory beetles. Within each of the volatile groups tested, subcortical beetles also showed differing responses to individual volatiles, and the MPB pheromone lure further altered their responses to them. We identified several key semiochemicals for each beetle family (Table 5). However, these results require further field testing before being implemented as semiochemical tools for monitoring and managing subcortical beetles.

In the NMDS analysis, we observed that MPB catches were highly associated with semiochemicals combining FVOCs and MPB pheromone lures (FVOCP). Notably, the FVOCP yielded the highest relative catches of MPB (86%) compared to the pheromone-alone (10%) and mineral oil control (1%). In contrast, FVOCs alone did not attract MPBs; there were no MPBs in traps containing individual FVOCs or the blend of FVOCs. However, in traps containing MPB lures, the 2-methyl-1-butanol, the 2-methyl-2-butanol, and the MPB lure alone (control) attracted 21%, 17%, and 11% of MPBs. These findings suggest that specific FVOCs associated with MPB symbiotic fungi, particularly 2-methyl-1-butanol and 2-methyl-2-butanol, may act synergistically with MPB pheromones to enhance MPB attraction. This aligns with the findings of Zaman et al. (2023b) [22], who demonstrated that MPBs are attracted to FVOCs released by their symbiotic fungi in laboratory bioassays. Therefore, these compounds may be useful semiochemicals when combined with MPB lure to attract low-density MPB populations in the field. However, further research is required to confirm their field efficacy.

The NMDS analysis also highlighted an association between predatory beetle families (Elateridae and Staphylinidae) and the SVC and healthy tree volatiles groups. The SVC experiment caught 30% of elaterids and 43% of staphylinids, compared to 10% and 12% in the mineral oil control, respectively. These results suggest that host volatiles may play a more significant role in attracting bark beetle predators than previously recognized. Although certain host tree volatiles, such as α-pinene, are known to enhance predatory beetle attraction to bark beetle pheromones [2,10,12,29], research on bark beetle predator attraction has focused largely on using bark beetle pheromones in combination with other volatiles in traps [71,72]. Therefore, our work represents the first attempt to investigate the role of SVCs in the attraction of bark beetle predators.

The attraction of both families of predatory beetles (Elateridae and Staphylinidae) was associated with the chemical profile of healthy trees. In the current study, the healthy tree profile caught 30% of elaterids and 21% of staphylinids compared to 12% and 15% in the control (mineral oil) treatment, respectively. Predators are known to utilize bark beetle pheromones and tree volatiles as semiochemical cues while foraging, especially tree volatiles indicating the presence of their prey [10,29]. Erbilgin et al. (2007) [2] proposed that tree-killing bark beetle species, like MPBs, are attracted to healthy trees since the thicker phloem of such trees increases MPB fitness [62,73]. Therefore, volatiles released by healthy trees may act as kairomones for predators by signaling ideal host conditions for prey species, such as MPBs. We speculate that individual monoterpenes like 3-carene within the healthy tree profile may act as an attractant for predators; coincidently, healthy trees have the highest concentrations of this monoterpene relative to the infected trees in this study. Furthermore, 3-carene attracts several bark beetle species, including the red turpentine beetle (*Dendroctonus valens*) [18,74]. Another monoterpene, α-pinene, could be responsible for the *E. harknessii* infected tree profile being identified as another key SVC profile for elaterid beetles. Miller (2023) [51] reported that predatory beetles of bark and wood-boring beetles were attracted to α-pinene, which is found in the highest concentration in trees infected by *E. harknessii* compared to other stress agents tested in the current study. The *E. harknessii* profile was also identified as a key SVC profile for both families of wood-boring beetles in this study. Therefore, we suspect α-pinene may indicate favourable hosts to wood-boring beetles, signalling their presence to predatory beetles, such as elaterids.

Cerambycids and buprestids, families of wood-boring beetles that compete with MPBs for resources in the host tree, interacted differently with each of the volatile treatment groups tested. Interestingly, buprestid catches were more strongly associated with semiochemical groups containing FVOCs, with 44% of buprestids caught in treatments with FVOCs and MPB pheromones (FVOCP) and 26% in FVOC-alone treatments, compared to only 12% in the mineral oil control. In contrast, cerambycids did not show any association with any particular volatile treatment group, with similar relative catches across FVOCP, FVOC, and SVC treatments. We found that relative catches in these volatile groups were consistently higher than in the volatile control groups (healthy tree, pheromone alone, and mineral oil controls). This suggest that a combination of all these volatile groups could be effective in attracting cerambycids.

In treatments without adding MPB lures, 2-methyl-1-butanol emerged as a key FVOC for the attraction of cerambycid and buprestid species caught in this study. These findings are consistent with previous studies that identified 2-methyl-1-butanol as an attractant for several wood-boring beetle species [44,45,46]. This compound was also identified as a key FVOC for MPB; however, this finding was observed only in treatments combining FVOCs with MPB lure. We observed that adding the MPB lure altered cerambycid attraction to 2-methyl-1-butanol relative to their attraction to the same compound alone. Furthermore, in treatments combining FVOCs with MPB lures, acetoin emerged as the primary FVOC attracting cerambycids, while isobutanol was the key FVOC for buprestids. Overall, these results highlight that wood-boring beetles interact differently with FVOCs in the presence of bark beetle pheromones. These responses further differ depending on the family of wood-boring beetles. These changes in the attractiveness of compounds in the presence of MPB pheromones must be considered when testing semiochemical tools for these species in the field.

We observed temporal variations in the catches of subcortical beetle communities. The highest catches of Scolytinae and Elateridae beetles occurred during the first collection (6–13 July), suggesting these beetles are most abundant early in the field season due to less competition for host trees. Similar trends in trap catches for both families throughout the season indicate closer interactions during host selection. A high catch of MPB during the last collection suggests two emergence periods of MPB in the 2023 field season. Higher catches of Cerambycidae and Staphylinidae beetles in the final collection indicate later summer emergence. Emergence times impact the semiochemical landscape and beetle behaviour, as beetles expend more energy flying in warmer temperatures, affecting dispersion, host colonization, and reproduction [75]. At our field sites, the hottest month was August, corresponding with our last two collection periods (3 August and 10 August). This could mean that cerambycids and staphylinids are more tolerant of hot temperatures during their adult flight period.

This study has some potential limitations. First, our sites were located in open areas with strong wind exposure. Several large wind events occurred during the field season. The high winds at the sites may have impacted the release rates and dispersal patterns of volatiles and pheromones, potentially altering beetle responses to our traps. However, these conditions likely affected all traps equally due to the weekly collections and trap rotations. Another potential limitation was the duration of our field season. We likely observed two separate emergence events of MPB at our field sites since our highest catches of MPB were in the first and last collections. Thus, we likely missed the MBP emergence at the beginning of the field season (before our first collection) and at the end of the second emergence period (after our second collection). In future experiments, extending the collection period from 5 weeks to 7 weeks may be more suitable, starting one week earlier and ending one week later, to ensure both populations are fully represented in the data.

We conclude that groups of subcortical beetle communities interact differently with various sources of volatiles. We identified 2-methyl-1-butanol as a key FVOC in synergizing MPB attraction to its commercially available lure. We also found that the addition of bark beetle pheromones to traps altered subcortical beetle community attraction to FVOCs, especially for wood-boring beetles. We do not know whether adding pheromones to SVCs would similarly alter beetle attraction to traps. Overall, our results suggest that FVOCs and SVCs could be useful semiochemical tools for monitoring and managing subcortical beetle species but require further field testing before being used in subcortical insect monitoring. Once developed, these tools will allow us to better understand beetle community composition and interactions between different functional groups of species.

## Figures and Tables

**Figure 1 insects-16-00057-f001:**
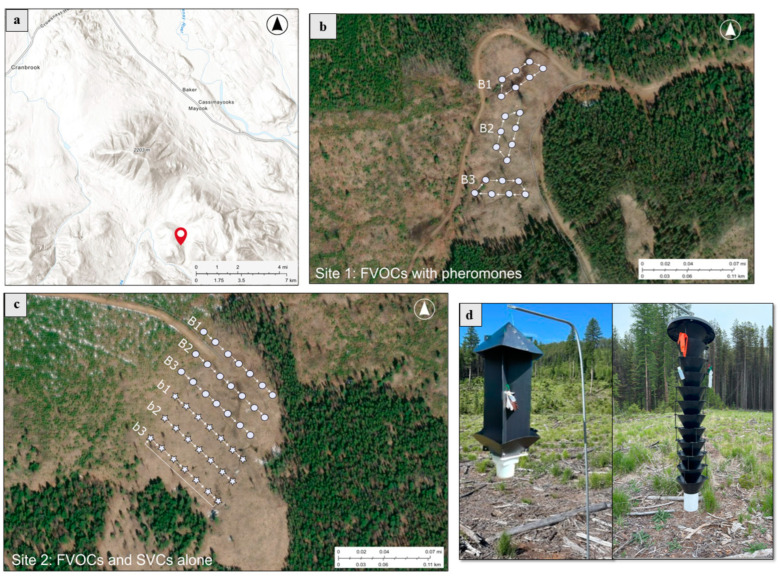
Experimental field site and trap setup. B1, B2, and B3 represent the three blocks of traps for each experiment, and arrows indicate the direction of trap rotation after each collection. (**a**) Location of sites within British Columbia. (**b**) Trap locations at field site 1; fungal volatiles (FVOCs) combined with pheromone experiment. (**c**) Trap locations at field site 2; FVOC-alone treatments (stars) and host stress volatiles (SVCs)-alone treatments (circles) were placed at this site. (**d**) Beetle traps used in our experiments: a flight intercept trap, used in our FVOC experiments (left) and a Lindgren funnel trap, used for the SVC experiment (right).

**Figure 2 insects-16-00057-f002:**
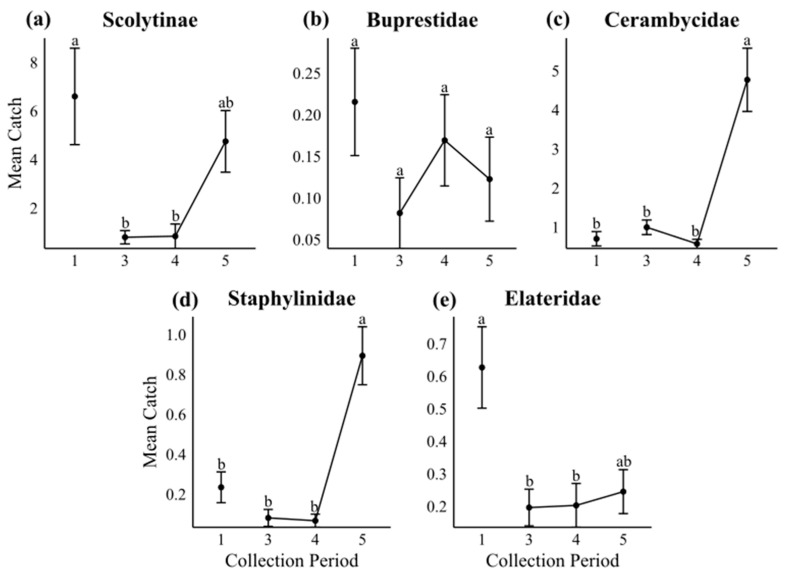
Mean catches of each subcortical beetle group throughout the collection period. Mean catches were calculated using data from all volatile groups. Traps were collected weekly for 5 weeks from 13 July (Collection 1) to 10 August (Collection 5). Catches from collection 2 are excluded due to the loss of traps during an extreme wind event between collections. Error bars represent standard error. Kruskal–Wallis tests were run for significant differences between collections at *p* < 0.05. Note: y-axes are on different scales due to large variations in catches between each subcortical beetle group. Collections with the same letter within each beetle group are not significantly different; Dunn’s test with Bonferroni’s adjustment at *p* < 0.05.

**Figure 3 insects-16-00057-f003:**
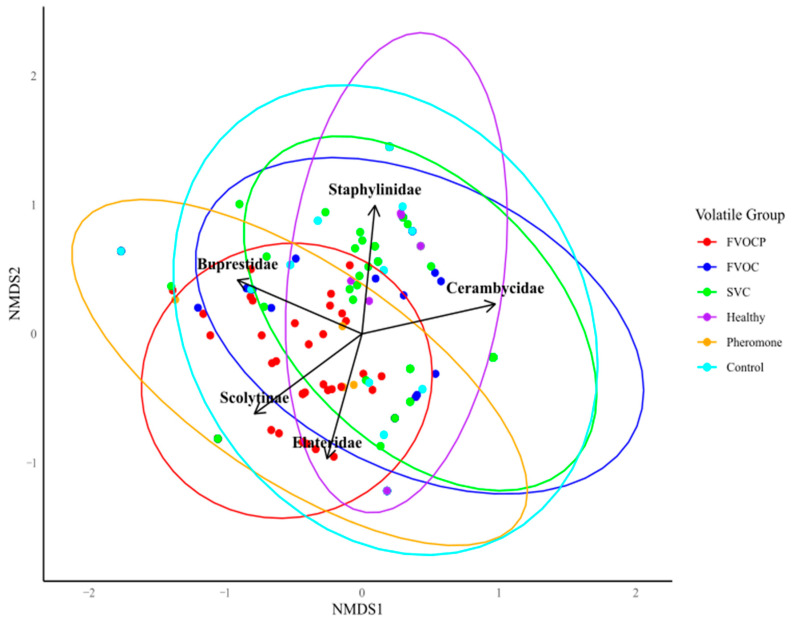
Non-metric Multidimensional Scaling (NMDS) plot showing the differences in subcortical beetle community composition between volatile treatment groups. Subcortical beetle catches over the entire field season were analyzed using an NMDS gradient analysis. Bray–Curtis was used as the distance metric. The vectors (black arrows) correspond to the five groups of subcortical beetles included in the analysis. The ellipses represent 95% confidence intervals around the centroid of each volatile treatment group. Volatile groups include (1) fungal volatiles combined with mountain pine beetle (MPB) pheromone lures (FVOCP), (2) fungal volatiles alone (FVOC), (3) host stress volatile chemical profiles associated with fungi (SVC), (4) healthy lodgepole pine volatile profile (Healthy), (5) MPB pheromone lures alone (Pheromone), (6) mineral oil alone (Control). Stress = 0.151, indicating goodness of fit for the ordination. Significant differences between volatile groups were determined by PERMANOVA using Bray–Curtis dissimilarity (at *p* < 0.05).

**Figure 4 insects-16-00057-f004:**
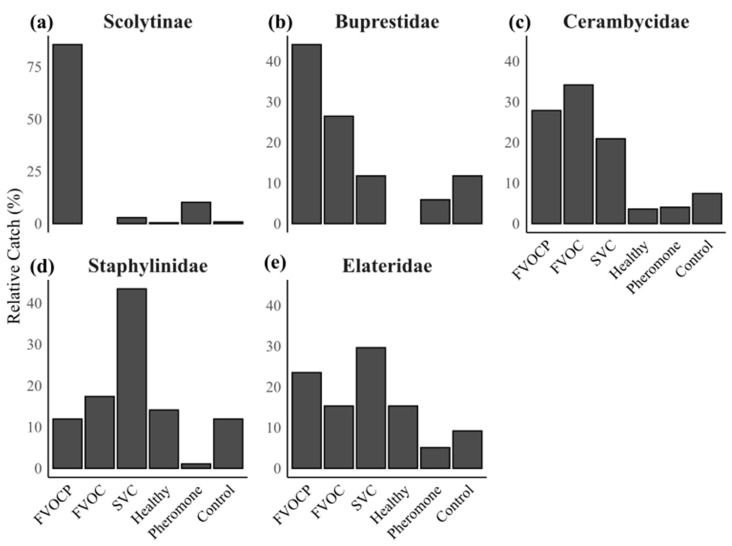
Trends in percent relative catches of subcortical beetle families between volatile treatment groups. Volatile treatment groups include: (1) fungal volatiles combined with mountain pine beetle (MPB) pheromone lures (FVOCP), (2) fungal volatiles alone (FVOC), (3) host stress volatile chemical profiles (SVC), (4) healthy lodgepole pine volatile profile (Healthy), (5) MPB pheromone lures alone (Pheromone), (6) mineral oil alone (Control). Note: the *y*-axis for (a) Scolytinae is on a different scale due to larger relative catches in one volatile treatment group compared to other subcortical beetle families.

**Table 1 insects-16-00057-t001:** Summary of known attractants (including pheromone components and host volatiles) for coleopteran species in the subcortical beetle families. Synthesized using Gitau et al. (2013) [30] and Tóth (2013) [31].

Subcortical Beetle Community	Known Attractants	Source	Species Attracted	References
Curculionidae, Scolytinae	*trans*-Verbenol	Female pheromone	*Dendroctonus* *ponderosae*	[32,33]
	*exo*-Brevicomin	Male pheromone	*D. ponderosae*, *D. terebrans*, *D. brevicomis*	[34]
	(-)-*endo*-Brevicomin	Male pheromone	*D. frontalis*	[35,36]
	(+)-Sulcatol	Female pheromone	*Gnathotrichus sulcatus;* *Ips sexdentatus*	[37]
	Frontalin	Female pheromone	*D. rufipennis*, *D. brevicomis*, *D. pseudotsugae*	[35,38,39]
	Ipsdienol	Male pheromone	*Ips* spp.	[26,40,41]
	α-Pinene	Host tree volatile	*D. frontalis*, *D. brevicomis*, *Ips* spp., *Hylastes porculus*	[18,32,42,43]
Cerambycidae	2-Methyl-1-butanol	Pheromone component	*Neoclytus*, *Xylotrechus* spp., *Monochamus* spp.	[44,45,46]
	Ethanol	Similar compound to male pheromone	*Monochamus* spp., *Xylotrechus* spp.	[43,47,48]
	Monochamol	Male aggregation pheromone	*Monochamus* spp.	[49]
	3-Hydroxy-2-hexanone	Male aggregation pheromone	Various Cerambycinae species	[46,50]
Buprestidae	α-Pinene	Monoterpene	*Buprestis lineata*	[23,43,51]
Staphylinidae	Isopropyl (Z9)-hexadecenoate	Female aggregation pheromone	*Aleochara curtula*	[52,53]
Elateridae	Ethanol	Released by stressed trees	*Alaus myops*	[43]
	Geranyl butanoate	Female aggregation pheromone	*Agriotes* spp.	[54,55]

**Table 2 insects-16-00057-t002:** Chemical concentrations and release rates of fungal volatile compounds (FVOCs) used in volatile dispensers. All chemicals were purchased from Sigma-Aldrich (St. Louis, MO, USA).

Treatments/Chemical	Chemical Purity (%)	Concentrations (µL mL^−1^)	Release Rate * (mg day^−1^)
Acetoin	≥96	61.87	1.48
3-Methyl-1-butanol	98	39.87	0.98
2-Methyl-1-butanol	≥99	20.92	0.56
Isobutanol	≥99	30.33	0.62
2-Methyl-2-butanol	99	33.27	0.84
FVOC mixture		48.39	0.96
Mineral oil			

* Release rates determined by weight lost in the laboratory at 22 °C.

**Table 3 insects-16-00057-t003:** Chemical purity of volatile compounds included in dispensers of host stress volatile (SVC) blends. Dispensers contained the same compounds mixed at different concentrations based on the SVC profile.

Compounds	Enantiomeric Ratios	Chemical Purity (%)	Source
Geranyl acetate		≥97	Sigma-Aldrich
α-Pinene	(−)	98	Sigma-Aldrich
Camphene	(−)	90	SAFC (Kent Town, Australia)
β-Pinene	(+)	≥94	TCI Chemicals (Tokyo, Japan)
3-Carene		90	Sigma-Aldrich
β-Myrcene		90	Sigma-Aldrich
Limonene	(S)-(−)	96	Fluka Analytical (Buchs, Switzerland)
Terpinolene		≥90	SAFC
Bornyl acetate		≥99	SAFC
γ-Terpinene		97	Fluka Analytical
Camphor		≥95	Fluka Analytical
Borneol	(−)	97	Sigma-Aldrich

**Table 4 insects-16-00057-t004:** Total catches of subcortical beetle families in field traps over the entire collection period in each volatile experiment. We tested three experiments: (1) fungal volatiles (FVOCs) alone, (2) FVOCs with the addition of mountain pine beetle (MPB) lures, and (3) host stress volatile (SVC) profiles. The SVC treatments include the profiles of lodgepole pine infected by *Atropellis piniphila* (AP), *Grosmannia clavigera* (GC), Healthy tree (Healthy), *Leptographium longiclavatum* (LL), *Ophiostoma montium* (OM), and *Endocronartium harknessii* (EH). In the SVC experiment, the healthy tree profile (Healthy) was used as a positive control and mineral oil alone as a negative control. The fungal volatile (FVOC) blend treatment (Mixture) refers to a synthetic blend of FVOCs based on the *O. montium* profile.

Group	Treatments	Scolytinae	Buprestidae	Cerambycidae	Staphylinidae	Elateridae
FVOC	2-Methyl-1-butanol	0	4	36	1	3
	2-Methyl-2-butanol	0	1	27	3	3
	3-Methyl-1-butanol	0	2	31	1	1
	Acetoin	0	1	17	3	3
	Isobutanol	0	1	20	4	2
	Control- Mineral oil	0	2	19	2	3
	Mixture	0	0	21	4	3
	Total	0	11	171	18	18
FVOC with pheromones	2-Methyl-1-butanol	148	2	15	4	3
	2-Methyl-2-butanol	118	2	8	1	7
	3-Methyl-1-butanol	83	2	6	2	5
	Acetoin	95	2	45	3	2
	Isobutanol	85	4	15	1	1
	Control- Pheromone alone	74	2	18	1	5
	Mixture	91	3	35	0	5
	Total	694	17	142	12	28
SVC	AP	5	1	17	9	7
	Control- Mineral oil	6	2	14	9	6
	GC	4	0	16	15	2
	Healthy	3	0	16	13	15
	LL	5	2	21	5	0
	OM	4	1	13	6	7
	EH	3	0	26	5	13
	Total	30	6	123	62	50
	Total Catches	724	34	436	92	96

**Table 5 insects-16-00057-t005:** Summary of known attractants for subcortical beetles, key fungal volatiles (FVOCs), and host stress volatile (SVC) profiles identified in each of the experiments in our study. The blend refers to a synthetic blend of FVOCs based on the *Ophiostoma montium* profile identified by Zaman et al., (2023b) [22].

Subcortical Beetle Family	Known Attractants	Key FVOCs	Key FVOCs When Pheromones Are Added	Key SVC Profiles
Curculionidae Sub-family: Scolytinae	*trans*-Verbenol, *exo*-Brevicomin, Terpinolene, Myrcene	Low catch	2-Methyl-1-butanol, 2-Methyl-2-butanol	Low catch
Cerambycidae	2-Methyl-1-butanol, ⍺-Pinene	2-Methyl-1-butanol	Acetoin, blend	*Endocronartium harknessii*
Buprestidae	Tree stress volatiles primarily	2-Methyl-1-butanol	Isobutanol	*Leptographium longiclavatum*
Staphylinidae	Bark beetle pheromones, some monoterpenes	Isobutanol, blend	2-Methyl-1-butanol, Acetoin	*Grosmannia clavigera*, Healthy tree
Elateridae	Bark beetle pheromones, some monoterpenes	Low catch	2-Methyl-2-butanol	Healthy tree, *Endocronartium harknessii*

## Data Availability

The authors will make the raw data supporting this article’s conclusions available upon request.

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
