# Peer review of "Navigating the Semiochemical Landscape: Attraction of Subcortical Beetle Communities to Bark Beetle Pheromones, Fungal and Host Tree Volatiles"

_insects, 2025, doi:10.3390/insects16010057_

Round 1
Reviewer 1 Report (Previous Reviewer 1)
Comments and Suggestions for Authors
The resubmitted the manuscript was well improved and can be accepted fir publication in Insects after minor revision:
- line 14 and others: insert a space between numbers and units
- line 197: which kind of ration? v/v?
- Table 3: release rates are not shown in the table. The two sentences should be shifted into the text
- Ref. 49: pinene in lowercase letters
- Ref. 53: Staphylinidae in uppercase letters
- Ref. 70: temperature in lowercase letters.
Author Response
Please see the attachment.

Reviewer 2 Report (New Reviewer)
Comments and Suggestions for Authors
The ms “Navigating the semiochemical landscape: attraction of subcortical beetle communities to bark beetle pheromones, fungal and host tree volatiles” is about forest beetle and semiochemcials. Basically, the ms. demonstrated much information about subcortical beetles and semiochemicals in the forest land and deserve our attention, and the results and discussion present useful information to further develop the attractant for different subcortical beetle, but there are some flaws in the experimental design and need the authors give more explanation or discussion. Here is some my suggestion:
Introduction:
Line 49-50: In the sentence “Species-specific pheromones are commonly used to monitor tools economically important bark beetle species” , “tools” should be deleted.
Materials and Methods:
1. In 2.1 Experimental design, why the authors did not test the synergistic effects of MPB pheromones and SVCs ? Also, FVOCs, SVCs and MPB pheromone together? Need explanation.
2. Line 143-144, the authors refer that “these traps were placed at a different location from the other two experiments to avoid possible spillover effects from the use of the MPB lure.”, but the premise was the population density of MPB should be similar between different experiment sites, please confirm this.
3. Line 146-148, and line 169-170, figure 1d, why used different types of traps? Your aim was to test the difference of chemical signals, so the authors should use the similar type of traps, or you could not compare them theoretically. Also in figure 1c, did the author consider about the edge effect of trap setting? Why not the author test FVOCs and SVCs with pair wise setting in experiment 2?
4. In Table 2, the author used μl/ml to describe the concentration, maybe the author better used mole quantities.
5. Line 189-190, line 193-194, and line 199, three traps for each treatment, I know this was a labor costing work in the mountain area, but three replicates may not enough.
6. Line 216: “mixed at different concentrations depending on the stress agent (Table 3, [16])”, I could not find any description in table 3. Line 217, what were the seven treatments? May be you should refer it to table 4.
7. In table 4, the description of SVC should be more detailed.
8. In figure 4, the X-axis title and category were lost in a) and b)
Round 2
Reviewer 2 Report (New Reviewer)
Comments and Suggestions for Authors
No further comments.
This manuscript is a resubmission of an earlier submission. The following is a list of the peer review reports and author responses from that submission.
Round 1
Reviewer 1 Report
Comments and Suggestions for Authors
Authors studied the attraction of five subcortical beetles (bark beetles and their competitors) by semiochemicals from stressed pine trees, three fungal symbionts, and a bark beetle pheromone in British Columbia. Such field studies have been done very often, but the landscape aspect in this study is quite novel.
However, I have serious concerns about a field study like this that was conducted in just one season (5 weeks)! Especially when the results are not very clear, as the author themselves say: "This study has potential limitations" or "These findings suggest that these chemicals warrant further field testing for potential use in monitoring and management of subcortical beetle populations".
I suggest that data from at least two to three seasons, with different bark beetle infestations and varying environmental conditions, should be presented. The catching period per year should be longer than 5 weeks.
Interesting that the response of bark beetle competitors to fungal and host stress volatiles changed in the presence of Dendroctonus ponderosae (MPB) lures. Unfortunately, authors do not explain the nature of the MPB lures. To say "a mix of exo-brevicomin, trans-verbenol, terpinolene, and myrcene" is not enough.
Results in Fig. 2 on the seasonal overall catches are not at all surprising. It is a pity that the results from catch 2 could not be taken into account due to extreme wind conditions.This shows even more the need of two or three experimental seasons.
The manuscript needs editorial and spelling corrections:
- Table 1 [55, 56] is incomplete
- line 104: et al.
- line 147: 15 mL
- line 178: I cannot find treatment informations in Table 2
- line 198: coleopteran families
- line 337: explain N/A
- Ref. 48 and others: why did authors write some journal titles in lowercase letters?
- a chapter "Conclusions" at the end of the text is desirable.
Comments on the Quality of English LanguageEnglish is fine.
Reviewer 2 Report
Comments and Suggestions for Authors
Authors tested the subcortical beetle attraction potentials of host plant volatiles, fungal volatiles, and mountain pine beetle pheromone. They caught various beetles using volatile traps in open areas surrounded by lodgepole pine forests and enumerated the catches. They inferred that the bark beetles were attracted by the combination of fungal volatiles and pine beetle pheromone. They also suggested that the predatory beetles were attracted by the host plant volatiles.
In the backdrop of growing overuse of hazardous synthetic pesticides, such semiochemical characterization works are important for discovering the non- or less hazardous biopesticides. They are also important for the plant-insect interaction chemical ecology studies. Following are the suggestions for the improvement.
Major suggestions
All the volatiles/ blends used in the study warrant detailed introductions. Authors have just cited previous works to establish that these are good candidates for this study. Details of whether these compounds were proved to be the actually emitted compounds, whether these were the only emitted ones or only identified ones must be included. Overall work assumes that FVOCs and SVCs contain only the compounds mentioned in this study. What if there were undetected and unidentified compounds in these blends?
Overall, there is an issue with the appropriate experimental control usage in this work, which needs to be seriously considered and solved.
· Were any positive controls used for different beetles to ensure that such volatile-mediated trapping is applicable for these beetles, in this setup? If yes, please clearly state the positive controls used in the experiments and explain the basis of their selections as positive controls. Please show how they were included in the analysis and how they impacted the results and conclusions.
· Trap setups with empty containers should also be included as negative controls.
· Line 173: “Our treatments also included… and a blank control (mineral oil).” Good that the blank, i.e., a negative control was used. However, the catches by the blank are not shown in Fig. 4. Please provide the data. Were they used in the analysis? If yes, were the results significantly different than the treatments?
· Line 182: “We used MPB lures alone as the control.” Was this a positive control? If yes, how were its results statistically different from the negative control and treatments?
· Line 206: “Key treatments for… to the control treatments.” These were positive or negative controls?
· Fig. 4 A shows significant effect of pheromone lure; however, it seems to work only for Scolytinae. “Pheromone alone” could have been used as a positive control for Scolytinae. What about the other beetles?
· As per table 4 and Fig. S1-S3, negative control (blank) and pheromone alone hardly showed notable catch differences when compared with various treatments.
· Was the blank included in the FVOCP experiment? If yes, please define.
· How the results of controls fared in the comparison with treatments?
· Line 349: “Catches of predatory beetles from Elateridae and Staphylinidae were highly correlated with SVCs…”. Please correct. As per Fig. 4e, SVC treatment’s Elateridae catches were not significantly higher than the other treatments. In fact, in absence of the comparisons with positive and negative controls, this cannot be inferred.
Fig. 3: Did the SVCs include “healthy”? Line 361: “Therefore, it is possible…signaling ideal host conditions for prey species, such as MPBs.” This is too speculative! To prove such an anticipatory behavior, is there a data, at least a beetle abundance data from the field? Did the Fig. 3 NMDS results differ with and without “healthy”? If yes, please provide the details.
Please correct the statistical significance denoting letters in Fig. 4A and 4C. In 4C, especially when standard error is plotted, can FVOC and FVOCP means be significantly different?
Lines 408 to 418: To avoid such limitations, often, first the behavioral assays coupled with electroantennography are conducted in controlled environments to validate activities of the candidate compounds and blends.
Please add the highest Y-axis values in Fig. 2 and 4. Please add the X-axis labels in Fig. 4.
Please explain how the release rates were determined. Table 3 legend states that “release rate of chemicals are 22.06 mg/day”. Did all volatiles have the same release rate?
Minor suggestions
Line 54: “synergistic” should be “synergists”.
Line 204: “Due to a wind event… excluded from all analyses.” This is redundant; already mentioned in line 155.
Line 225-226: “Of the 724 scolytids caught in our traps, only 14 were not MPB.” Please correct. MBPs are not scolytids.
English language improvement, typographical errors and brevity:
Line 41: “behaviour-modifying semiochemicals” can be just ‘semiochemicals’. Overemphasizing not necessary.
Line 54: “have explored”, Line 98: “have characterized” can be just “explored” and “characterized”; “have” can be avoided.
Line 110: “The two most common groups of SVCs are” can be “The two most common SVC groups are”
Line 132: MPB pheromone-mediated attraction? Or synergistic effect of FVOCs and MBP pheromones?
Line 53: Please add a full stop after “healthy trees [14,15,16]” and capitalize the “t” after it.
Table 1: Change Ip scallighraphus to Ips calligraphus. Alpha pinene is written in two different formats; please correct.
Table 2: Please change uL to µL.
Table 5: In the SVCs column, fungi names are written instead of chemical compounds; please correct.
“et al” or “et al.”? Please correct.
Line 161: Please reduce the space before “B1,…” .
Comments on the Quality of English LanguageThis manuscript needs English language editing. Brevity must be improved. There are several typographical errors; some examples are pointed in the review.
